# Allometric Equation for Aboveground Biomass Estimation of Mixed Mature Mangrove Forest

**Hazandy Abdul-Hamid** [1,2,*], **Fatin-Norliyana Mohamad-Ismail** [2,*], **Johar Mohamed** [1], **Zaiton Samdin** [2,3], **Rambod Abiri** [1,2], **Tuan-Marina Tuan-Ibrahim** [4], **Lydia-Suzieana Mohammad** [4], **Abdul-Majid Jalil** [2,5] **and Hamid-Reza Naji** [6]

1. Department of Forestry Science and Biodiversity, Faculty of Forestry and Environment, Universiti Putra Malaysia, UPM Serdang 43400, Selangor, Malaysia; johar.mohamed@upm.edu.my (J.M.); rambod.abiri@gmail.com (R.A.)
2. Laboratory of Bioresource Management, Institute of Tropical Forestry and Forest Products, Universiti Putra Malaysia, UPM Serdang 43400, Selangor, Malaysia; zaisa@upm.edu.my (Z.S.); majid@frim.gov.my (A.-M.J.)
3. Faculty of Economy and Management, Universiti Putra Malaysia, UPM Serdang 43400, Selangor, Malaysia
4. Forest Economics Section, Forestry Department Peninsular Malaysia, Headquarters, Jalan Sultan Salahuddin, Kuala Lumpur 50660, Federal Territory of Kuala Lumpur, Malaysia; marina@forestry.gov.my (T.-M.T.-I.); lydia@forestry.gov.my (L.-S.M.)
5. Natural Products Division, Forest Research Institute Malaysia, Kuala Lumpur 52109, Selangor, Malaysia
6. Department of Forest Science, Faculty of Agriculture, Ilam University, Ilam 69315-516, Iran; h.naji@ilam.ac.ir
* Correspondence: hazandy@upm.edu.my (H.A.-H.); fatinnorliyana22@gmail.com (F.-N.M.-I.)

**Abstract:** The disturbance of mangrove forests could affect climate regulation, hydrological cycles, biodiversity, and many other unique ecological functions and services. Proper biomass estimation and carbon storage potential are needed to improve forest reference on biomass accumulation. The establishment of a site-specific allometric equation is crucial to avert destructive sampling in future biomass estimation. This study aimed to develop a site-specific allometric equation for biomass estimation of a mix-mature mangrove forest at Sungai Pulai Forest Reserve, Johor. A stratified line transect was set up and a total of 1000 standing trees encompassing seven mangrove tree species were inventoried. Destructive sampling was conducted using the selective random sampling method on 15 standing trees. Five allometric equations were derived by using diameter at breast height (D), stem height (H), and wood density (ρ) which were then compared to the common equation. Simulations of each allometric equation regarding species were performed on 1000 standing trees. Results showed that the single variable (D) equation provided an accurate estimation, which was slightly improved when incorporated with the H variable. Both D and H variables, however, gave inconsistent results for large-scale data and imbalance of sampled species. Meanwhile, the best fit either for small-scale or large-scale data, as well as for imbalanced sample species was achieved following the inclusion of the ρ variable when developing the equation. Hence, excluding the H variable while including the ρ variable should be considered as an important determinant in mixed mangrove species and uneven-aged stand for aboveground biomass estimation. This valuation can both improve and influence decision-making in forest development and conservation.

**Keywords:** mangrove; aboveground biomass; tree component; allometric equation; power function

## 1. Introduction

Mangrove forests play unique ecological functions in subtropical coastal regions [1–4]. It takes years for the ecosystem to reach the maturity phase to facilitate the provision of providing essential services, such as fisheries, timber and fuelwood production [5,6], habitat protection [7], coastal defense [8], and carbon sink production in the tropics [9–11]. In compliance with the Reduced Emission from Deforestation and Land Degradation

(REDD+) [12,13], the Malaysian Government must provide the national Forest Reference Level of mangrove forest biomass productivity and carbon stock to the United Nations Framework Convention on Climate Change (UNFCC). The proper and accurate estimation of forest biomass is one of the 22 elements to determine the Total Economic Valuation (TEV) for forest ecosystems and services based on the framework by The Economics of Ecosystem and Biodiversity (TEEB) [14] as explained by De Groot et al. [15].

The mangrove forest geomorphological condition is limited in terms of accessibility, time consumption, and in posing a threat to worker's safety. In relation to the tidal water, mangrove root systems such as the numerous massive stilt roots, knee roots and pneumatophores systems that outgrow the trunk and grow vertically above the soil, these may threaten worker's safety. Nevertheless, the aboveground biomass of mangrove forests still needs to be estimated, considering their exceptional roles and services to the environment.

There is a growing interest in estimating forest composition by using Unmanned Aerial Vehicles (UAVs), such as drones and remote sensing in capturing forest imagery based on forest canopy that can be further described based on color features. The use of UAVs is advantageous, especially in forest areas that are difficult to access. Additionally, the use of drones and remote sensing is vital in managing and monitoring forests from undesirable practices, such as illegal logging. However, UAVs are unreliable is estimating certain crucial data for determining forest carbon stock. Examples of these data include wood density and biomass by tree component (trunk, branch, leaves, flowers, and fruits) that support the aboveground biomass accumulation. On the other hand, different definitions of forest canopy height (such as the mean height of all trees, basal-area weighted height, or height of the tallest tree within a certain area) that are generated from airborne lasers/Lidar (ALS), lead to contracting results from the same datasets, [16–18] as regular inundation might result in an error in the height retrievals [19].

Mathematically, various methods have been developed by forest ecologists in estimating forest biomass, involving the relationship between the biomass of whole trees and, their components, as well as some readily measured parameters such as the diameter of the stem at breast height (DBH), stem height (H), and wood density ($\rho$). One of the methods is by developing allometric equations with destructive biomass, which requires a small number of tree samples to be harvested and the estimation may be performed by either the whole or partial tree weight from the measurable tree dimension. This method was preferred instead of the destructive and mean-tree methods that require the harvest of all the trees in developing the equation [11,20–25].

Allometric relationships between the aboveground biomass and the DBH parameter have been reported for specific mangrove species, such as *Rhizophora apiculata* (L.) Blume [26,27] and *Bruguiera parviflora* (R.) Wight [26]. The present study focused on determining the biomass accumulation on a natural mature mangrove forest occupied by mixed species. This study will improve the current knowledge on forest conditions facing other land areas (Singapore) or mangrove islands, where possibly seed sources are received from the nearest forest, rather than the areas directly facing the vast ocean of rough tides, huge waves, strong winds, and tropical storms, such as typhoons and hurricanes. By using primary data, the verified single developed equation (with several relationships) that fits the available mangrove species could help in efficiently managing forest. It will also provide the relevant figures of forest canopy layers and forest profiles regarding biomass accumulation and carbon stock at this forest area, instead of applying single species models that may require higher cost and are time consuming in data inventory, gathering, and presenting.

Nonetheless, in ensuring the validity of the equation that will be developed, there is a need to have an existing equation to rely on by considering the geographical origin and species that make up the data set of the derived equation. A previous study conducted by Hazandy et al. [28] developed allometric equations for estimating aboveground biomass in the Matang Forest Reserve (northern part of Peninsular Malaysia), however, the researchers found the equation to be less suitable for application in the present study. Moreover, the equation developed by Hazandy et al. [28] focused on the even-age planted mangrove.

In this study, the equation developed by Komiyama et al. [11] known as 'the common equation' was applied for both practical and comparison application due to the segregation of the species and the similarity of the study site conditions to that of the Asian region.

Apart from the D and H, the wood density (ρ) is one of the important enablers in estimating aboveground biomass as it differs significantly among various mangroves species [21]. A lower difference of ρ is only found for various individuals within a species [29]. This study aimed to develop a site-specific allometric equation by considering tree wood density ρ in relationship to DBH and H for a mixed mature mangrove forest in the Sungai Pulai Forest Reserve in the southern part of the Peninsular Malaysia.

## 2. Materials and Methods

### 2.1. Sampling

The site for the study is located at Sungai Pulai Forest Reserve; in the southeast of Pontian and Johor Bahru district (01°27′ N, 103°33′ E). It is the largest mangrove forest in Johor state and the second largest in Peninsular Malaysia. Ground-truthing (visual assessment) was conducted to identify high, medium, and low standing tree distribution to ensure that the range of biomass is sampled [30,31]. The transect line was set up across Compartments 16, 259A, 412B, and 453B Sungai Pulai Forest Reserve, Johor (Figure 1). Plot establishments were performed near the river or the estuary. For each compartment, two plot designs of 50 × 50 m each (total plot area = 2 hectares) were randomly established from the marine to the center of the compartment [31]. Parameters such as D and H were measured for trees of 5 cm diameter and above, and the species were identified. In the total of 2 hectares (ha), a total of 1386 standing trees were inventoried in the plotted area of 2 hectares (ha). Thereafter, 1000 of the standing trees were randomly selected to derive the perform equation.

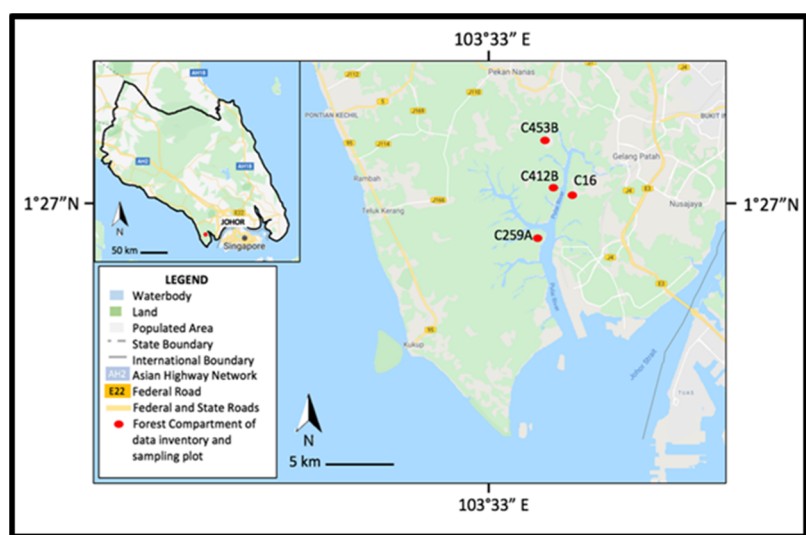

**Figure 1.** Study area in the Sungai Pulai Forest Reserve showing the forest compartment of the data inventory sampling plot.

The D was measured using the Forestry Suppliers Metric Fabric Diameter Tape (Model 283D/10M, Jackson, MS, USA), whereas the H was measured using the Suunto Compass Clinometer (Model PM-5, Tammiston Kauppatie, Vaanta, Findland). Main stem tree diameter was measured at diameter breast height, approximately 1.37 m above the ground. Stem diameter is often measured above the highest stilt root for stilt rooted trees such as *Rhizophora* spp. [29,32]. Two-compartments (Compartment 16 and 259A) were then selected to perform the destructive sampling by considering the suitability of the accuracy of biomass distribution, sampling time, and cost efficiency [31]. A selective random sampling method was used in selecting 15 standing trees that represented seven mangrove

species (Table 1) based on the normal distribution of diameter at breast height, stem height (Figure A1 in Appendix B), and species composition.

**Table 1.** The number of sampled tree species and range of the diameter and height.

| Species | Local Name | Sample Trees | D (cm) | H (m) | Tree Density |
|---|---|---|---|---|---|
| *Bruguiera cylindrica* (L.) Blume | Bakau putih | 3 | $17.13 \pm 3.85$ | $17.17 \pm 1.23$ | $715.62 \pm 10.2$ |
| *Bruguiera parviflora* (R.) Wight | Lenggadai | 1 | $18.7 \pm 0$ | $18.0 \pm 0$ | $736.81 \pm 0$ |
| *Bruguiera sexangula* (L.) Poir | Tumu putih | 2 | $24.9 \pm 1.10$ | $20.50 \pm 0.50$ | $713.72 \pm 7.87$ |
| *Ceriops tagal* (P.) Rob | Tengar | 1 | $15.5 \pm 0$ | $14.5 \pm 0$ | $745.45 \pm 0$ |
| *Rhizophora apiculata* (L.) Blume | Bakau minyak | 5 | $20.08 \pm 4.21$ | $19.92 \pm 1.69$ | $842.22 \pm 28.5$ |
| *Rhizophora mucronata* (L.) Lam | Bakau kurap | 1 | $16.8 \pm 0$ | $18.1 \pm 0$ | $801.41 \pm 0$ |
| *Xylocarpus granatum* (J.) Koenig | Nyireh bunga | 2 | $14.70 \pm 4.10$ | $13.88 \pm 2.63$ | $708.65 \pm 17.9$ |

*2.2. Sample Preparation*

The fresh weight of each component (trunk, branches, and leaves) from the 15 destructive trees was measured in situ using the Brecknell 235 10X 300 kg mechanical hanging scale. For all trunk sample trees, the trunk diameters were measured at the lower, middle, and upper part of the log that was cut from ground level in the following lengths: (0–0.3 m, 0.3–1.3 m, 3.3–5.3 m and followed then for every 2 m lengths) [29]. The log was cut into several cutting logs—modified from Doruska et al. [33] to avoid exceeding the load of the weight balance as the trunk was assumed to be conical in shape [29] and to standardize the cutting log length for each tree. The lower part from each log sample was cut into a disc form (2 cm to 3 cm thickness) to obtain the range of $\rho$ because the variation of $\rho$ in individual species correlates with carbon allocation [34] and effective vertical stem expansion [35]. This could reduce the error with the different scaling factors among vertical log positions such as butt log, middle log, and upper log [33]. Thereafter, 500 g from the fresh weight of branch, twig, and leaf components were taken to the laboratory [29]. The samples were then oven-dried for 15 days at 70 °C until a constant weight was attained to calculate the dry weight conversions for each component [36]. The $\rho$ of the wooden disc (mass contained in a unit volume) [37] was determined from the stem and the largest branches [28].

*2.3. Approach in Biomass Estimation*

The biomass of each component was calculated by $B = M_{Fre} \times (M_{sam,Dry}/M_{sam,Fre})$, where $M_{Fre}$ is the fresh mass of each component and $M_{sam,Dry}$ and $M_{sam,Fre}$ are the dry mass and fresh mass of the samples of each component, respectively. The total aboveground biomass was obtained by summing the biomasses of each component.

The data were fitted to a non-linear regression model in the form of an intrinsically linear model of the power function, whereas the accuracy of the calculation was based on the coefficient of determination (R). The equation was derived as a single parameter by combining the diameters at breast height, stem height, and wood density to determine the variability explained by the model [38]. The equations were simplified as in the model shown below:

$$M = aD^b e \tag{1}$$

$$M = aD^2 H^b e \tag{2}$$

where,

M = Aboveground biomass
D = Diameter at breast height
H = Stem height
a and b = constant
e = error term

The allometric relationship of Equation (3) in a linear form was derived by taking the natural logarithms of both sides of the equation [38]:

$$\ln M = \ln a + b \ln \rho + b \ln D + \ln e \tag{3}$$

where,

$\rho$ = wood density

In this form, linear regression can be used to build the regression model that fits the biomass data. The calculated result is aimed at presenting the known values (common equation), whereas the percent error formula is used to determine the precision of the calculations. The experimental value is the calculated value while, the theoretical values are the common equation value. All the parameters used to develop the equation are influenced by measurement error, notably having different effects on the model parameters [39]. Meanwhile, the absolute error is the magnitude of the difference between the actual value and the estimated value. According to Bellasen and Stephan [40], an error value is acceptable if it is lower than 10% at a 90% confidence interval with an uncertainty factor of 1.5%.

The formula of the percentage of error is presented below:

$$\left[ \frac{(\text{Theoretical} - \text{Experimental})}{\text{Theoretical}} \right] \times 100 \tag{4}$$

where,

Theoretical = Known value
Experimental = Calculated value

### 2.4. Statistical Analysis

Further, the allometric equations of 15 destructive samples were derived using the General Linear Model, IBM SPSS statistic software version 25.0, IBM Corp, Armonk, NY, USA using the following: diameter at D, H, and $\rho$ variables, the aboveground biomass (kg) comparison between the observed value and predicted value for the equation using D, $D^2H$ and $\rho$ variables, homoscedasticity of residuals, and the developed equations of aboveground biomass (kg) estimation regarding species of 1000 trees in a 1 ha plot inventory of the Sungai Pulai Forest Reserve. The developed equations of aboveground biomass (kg) estimation regarding species of 1000 trees in the 1 ha plot inventory of Sungai Pulai Forest Reserve were derived using SigmaPlot Version 12.5, Systat Software, Inc., San Jose, CA, USA.

### 3. Results and Discussion

#### 3.1. Allometric Equations Derived from 15 Destructive Trees

The allometric equation derived for the relationship between D, H, and $\rho$ variables from 15 destructive sampling trees of seven species was in the range of DBH from 9.0 cm to 33.0 cm. Equations (1) and (2) were derived for each part of the tree samples (mass of stem, leaves, branch, and twigs) in a single D variable and a combination with the H variable (Table 2). The combination of the variables in the second equation was derived to study the level or degree of variability explained in the biomass accumulation on the stem, leaves, branch, and twigs. The equation derived is accepted to be normal distributed as the standardized and unstandardized residual normality were of the same value and greater than 0.05 (Table 2). Regardless of species, the results in Table 2 indicate that the $R^2$ values

were in the range 0.5834 to 0.9543 for Equation (1) (single D variable) and from 0.553 to 0.9556 for Equation (2) (incorporating the D and H variables).

**Table 2.** Summary of the allometric equation derived from 15 destructive samples using diameter at breast height, stem height, and wood density variables.

| Model | No. of Equation | Component | Equation | $R^2$ | Standardized and Unstandardized Residual Normality |
|---|---|---|---|---|---|
| $M = aD^b$ | 1 | (a) M of stem | $0.1761D^{2.3769}$ | 0.9223 | 0.107 |
| | | (b) M of branches and twigs | $0.0553D^{2.3055}$ | 0.7792 | 0.499 |
| | | (c) M of leaves | $0.0347D^{1.9762}$ | 0.5834 | 0.206 |
| $M = aD^2H^b$ | 2 | (a) M of stem | $0.0355D^2H^{0.9778}$ | 0.9579 | 0.833 |
| | | (b) M of branches and twigs | $0.0221D^2H^{0.8745}$ | 0.6880 | 0.853 |
| | | (c) M of leaves | $0.0125D^2H^{0.7767}$ | 0.5530 | 0.246 |
| $M = aD^b$ | 1 | Total biomass | $0.2999D^{2.3001}$ | 0.9543 | 0.117 |
| $M = aD^2H^b$ | 2 | Total biomass | $0.0739D^2H^{0.9291}$ | 0.9556 | 0.813 |
| $M = a\rho^{b1}D^{b2}$ | 3 | Total biomass | $0.00475\rho^{0.6309}D^{2.28787}$ | 0.9697 | 0.724 |

Note: M = biomass; D = diameter at breast height; H = stem height; ρ = wood density; a and b = constant.

The results revealed that the biomass of different tree components could be estimated using power equations based on a single D variable (stem $R^2 = 0.9223$; branch $R^2 = 0.7792$; and leaves $R^2 = 0.5834$) and the combination of D and H variables (stem $R^2 = 0.9579$; branch $R^2 = 0.688$; and leaves $R^2 = 0.553$). Equations (1) and (2) explained the aboveground biomass accumulation in tree components, in which the stem biomass was allocated the biggest biomass, followed by branches and twigs. This finding depicts that by either ignoring or excluding branch and twig samples during destructive sampling, the resulting equation becomes invalid. Meanwhile, the lowest $R^2$ value for both Equations (1) and (2) (Table 2) was from the leaf component, showing that crown distribution plays a minimum role in biomass allocation as highlighted by Komiyama et al. [11]. The exclusion of leaf biomass might be considered, however, the development of the equation remains inaccurate.

The equations that incorporated the H variable (Equation (2)) were found to be slightly higher compared to the equations using a single variable, D (Equation (1)). Meanwhile, the equation that incorporated both D and ρ variables (Equation (3)) resulted in the highest $R^2$ value (0.9697) compared to the equation that incorporated both the D and H variables, and the single D variable. Independent *t*-test (Table 3) also indicates that the allometric equations derived were statistically significantly different. Thus, the inclusion of the ρ variable to develop an allometric equation is not significant for aboveground biomass estimation that involves low species variation (planted forest) [36], nonetheless, the ρ variable must be considered in the biomass estimation of a variety of species, especially for uneven age mangrove forest [32,41,42].

**Table 3.** Summary of the independent samples t-test of the allometric equation derived from 15 destructive samples using diameter at breast height, stem height, and wood density variables.

| No. of Equation | | N | Mean | SD | SE | *t*-Value | *p*-Value | 95% Lower Bound | 95% Upper Bound |
|---|---|---|---|---|---|---|---|---|---|
| 1 | (a) | 15 | 5.0923 | 0.8731 | 0.1914 | 12.4186 | $1.3858 \times 10^{-8}$ | 1.9634 | 2.7904 |
| | (b) | 15 | 3.7284 | 0.8470 | 0.3404 | 6.7726 | $1.3161 \times 10^{-5}$ | 1.5701 | 3.0410 |
| | (c) | 15 | 2.3163 | 0.7260 | 0.4632 | 4.2667 | 0.0009 | 0.9756 | 2.9768 |
| 2 | (a) | 15 | 5.0923 | 0.8898 | 0.0569 | 17.1905 | $2.5414 \times 10^{-10}$ | 0.8549 | 1.1007 |
| | (b) | 15 | 3.7293 | 0.7942 | 0.1636 | 5.3369 | 0.0001 | 0.5216 | 1.2274 |
| | (c) | 15 | 2.3163 | 0.7068 | 0.1937 | 4.0105 | 0.0015 | 0.3583 | 1.1951 |
| 1 | | 15 | 5.4042 | 0.8449 | 0.1395 | 16.4836 | $4.2895 \times 10^{-10}$ | 1.9986 | 2.6015 |
| 2 | | 15 | 5.4042 | 0.8455 | 0.0555 | 16.7277 | $3.572 \times 10^{-10}$ | 0.8091 | 1.0491 |
| 3 | | 15 | 5.4042 | 0.8517 | 0.1184; 0.2555 | 19.3310; 2.4701 | $2.0743 \times 10^{-10}$; 0.0295 | 1.9838; −1.2527 | 1.6592; 2.6144 |

The scatter plot (Figure 2) of predicted value and residual value shows how much of an error the regression equation made with respect to predicting individual values in the dataset. The result indicates that the distribution of the residual values (dependent variable) is distributed uniformly and does not have any clusters forming together. The average proportions of biomass allocation of 15 destructive samples were 74% in stems, 20% in branches, and 6% in leaves. This result is consistent with the study conducted by Gong and Ong [43] on other mangrove forests in Peninsular Malaysia. Biomass accumulation of standing trees is relatively higher for the structural tissue and lower for the leaves [26,44–46]. This is due to a gradual increase in the absolute mass of stem and branches while the absolute mass of leaves tends to stabilize or be shed as litter upon attaining a certain tree size [47–49].

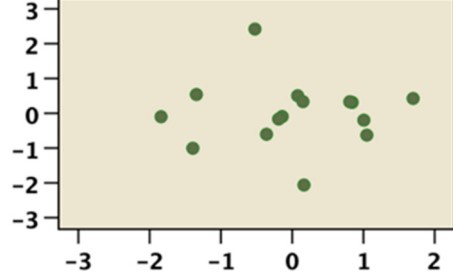

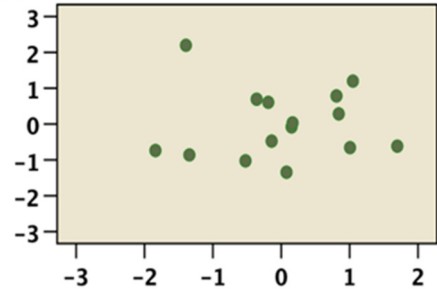

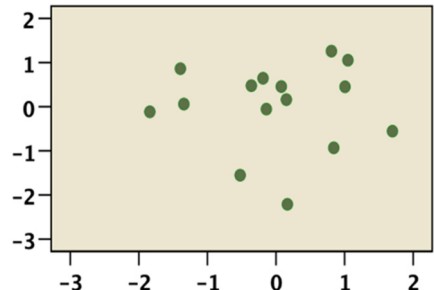

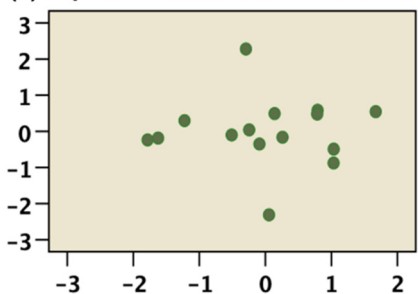

**Figure 2.** *Cont.*

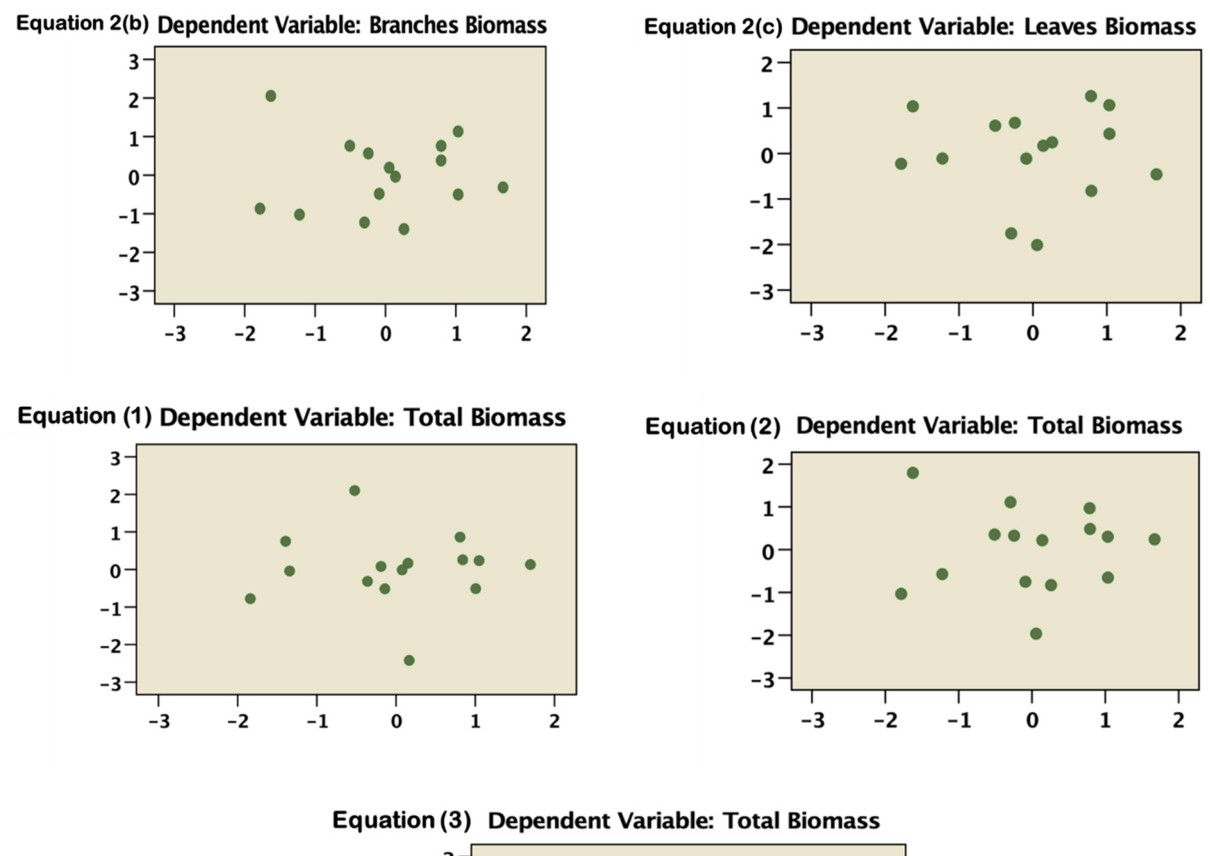

**Figure 2.** The test of homoscedasticity of residuals.

Five developed equations of 15 destructive trees were then regressed between the observed value (total biomass collected from the field) and the predicted value (D, H, and ρ value that were applied to the developed equation) (Figure 3). Regardless of species, the $R^2$ single variables (Equation (1)) were in the range of 0.9717 to 0.9718 and were found to be slightly increased (from 0.9717 to 0.972) for incorporated variables (Equation (2)). Meanwhile, the estimation of total aboveground biomass was similar upon applying the single D variable (Equation (1)) and incorporated variables (Equation (2)).

These two estimations (observed versus predicted) indicate the presence of a strong relationship between the tree variables (D and $D^2H$) for mixed species and uneven age of mangrove stand. Smith and Whelan [50] also found a good relationship between stem height and tree biomass in Florida mangroves and similar results were obtain when the diameter variable was used. Henry and Aarssen [51] reviewed the regressions between stem diameter and height and reported a lack of uniformity that might be influenced by biomechanical constraints and near neighbor effects. Comparatively, Equation (3) (0.9760) revealed that small destructive sampling (15 trees) with the inclusion of ρ variable provided a reliable and validated equation instead of the common equation ($R^2 = 0.9730$) (Figure 3) developed by Komiyama et al. [29] that used 104 destructive sampling trees (Appendix A).

To ascertain if the developed equation could fit large-scale data [51], the equation was applied to the primary data of 1000 standing trees.

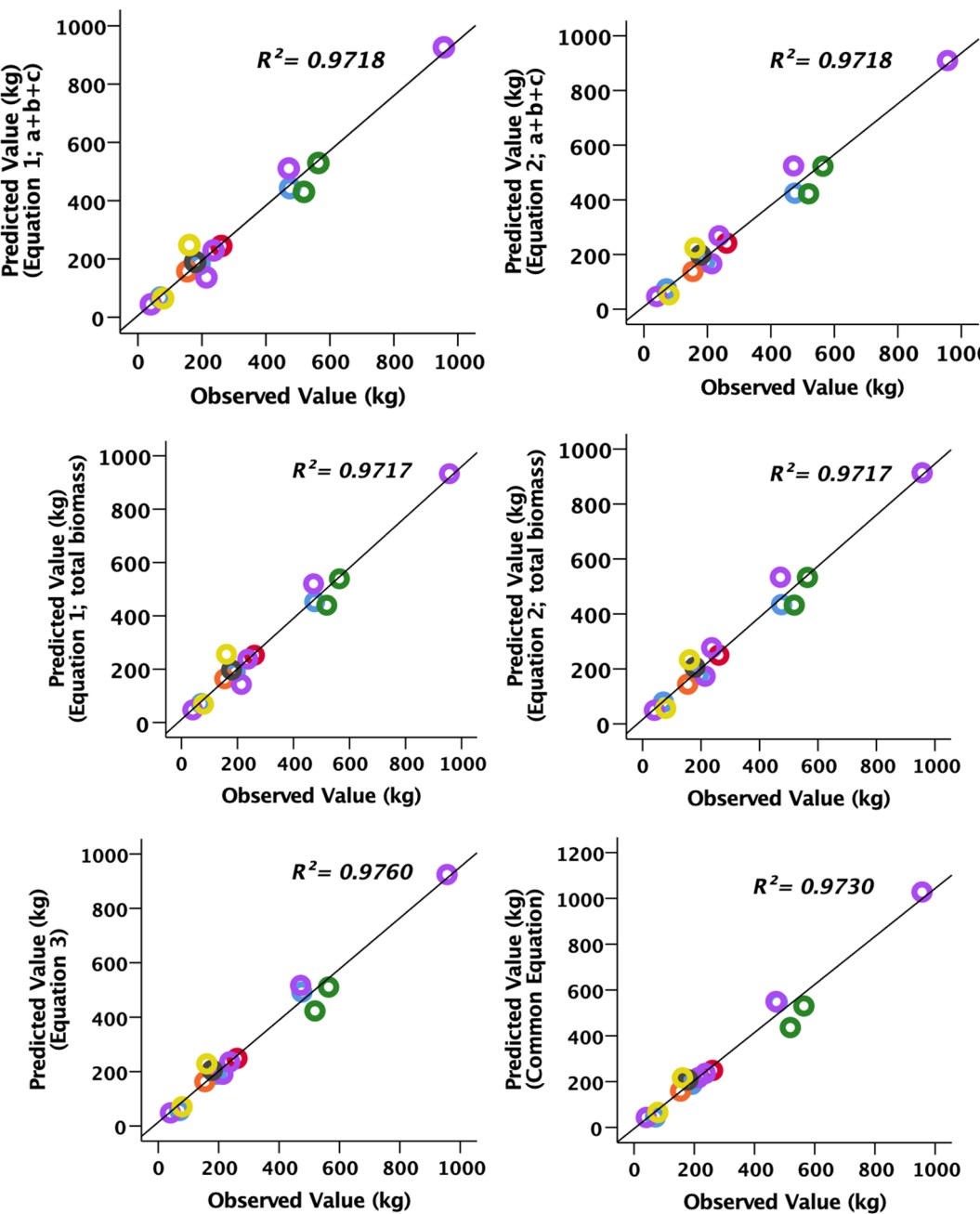

**Figure 3.** Comparison of the observed value and predicted value of aboveground biomass (kg) for a single variable (D) and incorporated variables ($D^2H$, and D and ρ variables) of 15 destructive trees. Blue: *Bruguiera cylindrica* (L.) Blume, red: *Bruguiera parviflora* (R.) Wight; green: *Bruguiera sexangula* (L.) Poir; orange: *Ceriops tagal* (P.) Rob; purple: *Rhizophora apiculata* (L.) Blume; dark grey: *Rhizophora mucronata* (L.) Lam; and yellow: *Xylocarpus granatum* (J.) Koenig.

### 3.2. Allometric Equations Applied to 1000 Standing Trees

Primary data inventory of seven species with a total of 1000 standing trees in Sungai Pulai Forest Reserve are listed in Table 4 with normal distributions of DBH and H as in Appendix B. Regardless of species, the range of D of 1000 standing trees in the study site was smaller but was still in the range of those used by Komiyama et al. [29] to develop a common equation (Appendix A). The reason for the limited size in the study site could be due

to the regeneration process of an uneven-aged stand from the previous harvest of charcoal production [52]. The *R. apiculata* species (553 individual trees) dominated a total of 1 ha area, followed by *Bruguiera cylindrica* (L.) Blume (185 trees), *Bruguiera sexangula* (L.) Poir (131 trees), *Rhizophora mucronata* (L.) Lam (44 trees), *B. parviflora* (40 trees), *Xylocarpus granatum* (J.) Koenig (39 trees), and *Ceriops tagal* (P.) Rob (8 trees). Species distribution in the study site revealed that *R. apiculata* accounted for 65.5% of the 1000 individual trees (Table 3).

**Table 4.** Summary of descriptive statistics for aboveground biomass 1000 standing trees in Sg. Pulai Forest Reserve.

| Species | No. of Tree | Diameter (cm) | | | Height (m) | | |
|---|---|---|---|---|---|---|---|
| | | Min | Max | Mean | Min | Max | Mean |
| *B. cylindrica* | 185 | 5 | 31.5 | [1] 15.09 ± 10.38 | 4.2 | 30 | 15.99 ± 0.33 |
| *B. parviflora* | 40 | 6.9 | 34 | 18.05 ± 1.13 | 6.5 | 28 | 18.75 ± 0.77 |
| *B. sexangula* | 131 | 6.1 | 34 | 16.35 ± 0.39 | 4.1 | 28 | 17.25 ± 0.30 |
| *C. tagal* | 8 | 13 | 25.2 | 19.45 ± 1.40 | 15 | 26 | 19.32 ± 1.26 |
| *R. apiculata* | 553 | 9.9 | 40.5 | 20.03 ± 0.24 | 10 | 35 | 19.51 ± 0.16 |
| *R. mucronata* | 44 | 11 | 27.8 | 17.38 ± 0.65 | 11 | 24 | 18.16 ± 0.42 |
| *X. granatum* | 39 | [2] 2.4 | 43.2 | 19.74 ± 1.21 | 5.2 | 24 | 14.83 ± 0.71 |

Note: [1] ± represent standard error, [2] The minimum value for *X. granatum* is below 5 cm diameter because the data were obtained from multiple leader trees.

A bigger dimensional size (mean D = 20 cm; mean H = 19.5 m) was also found from *R. apiculata* species. Likewise, previous studies showed that *R. apiculata* was the dominant species in other mangrove forests in Peninsular Malaysia and East Malaysia [26,53–56]. *B. cylindrica*, on the other hand, was documented to have the smallest dimensional size (mean D = 15 cm; mean H = 15.9 m) despite being the second-highest in the total number of individual trees. Plot establishments were performed near to the river, and the estuary because species distribution may be dominant for the seaward zone and mid-zone of mangrove species. However, it yielded an unbiased effect to the development of the equation due to the un-even age of the natural regeneration mix species. Moreover, the number of destructive samples on each species was based on the number of trees of each species in the primary inventory.

To perform in-depth study, to ascertain the internal consistency of the developed equation, the equation was applied to the primary data of 1000 standing trees regardless of species and in comparison, to the common equation (Figure 4). The best fit was found for Equation (3) with an $R^2$ value of 0.9979 and the lowest percentage error of 5.93% (Figure 4). The fitting ($R^2$ value) for both Equation (1) ($R^2$ = 0.9892) and Equation (3) ($R^2$ = 0.9888) were slightly lower than the Equation (3) with corresponding percentages error of 10.06% and 9.94%, respectively. The inclusion of the H variable for Equation (2) yielded the lowest data fit ($R^2$ = 0.9205 and 0.9196 respectively) and the highest percentage error (15.42% and 15.62% respectively). Haase and Haase [57], Rayachhetry et al. [58], and Chave et al. [41] reported that it is not advisable to include the H variable for equation development. Furthermore, Novitzky [59] found that the H increases with an increase in temperature and precipitation as the latitude decreases. On the other hand, Kodikara et al. [60] stressed that the effect varied with the soil salinity.

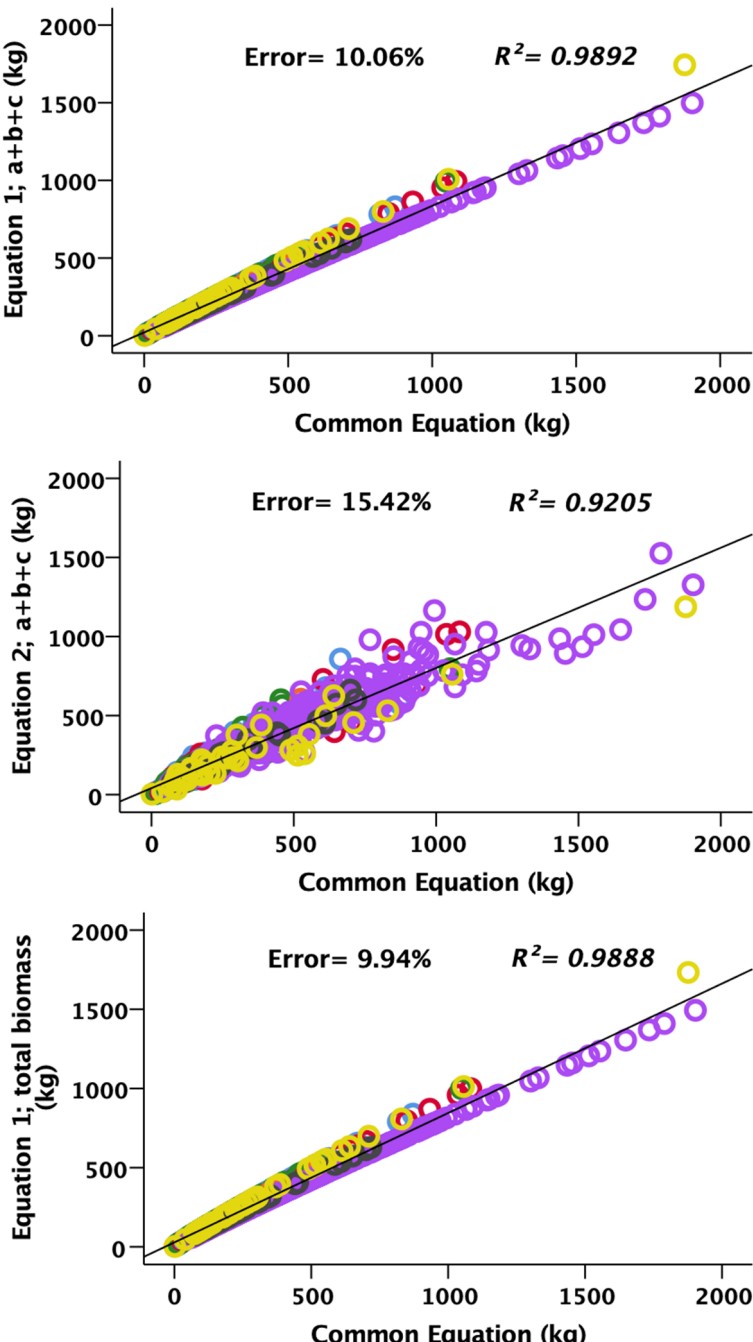

**Figure 4.** *Cont.*

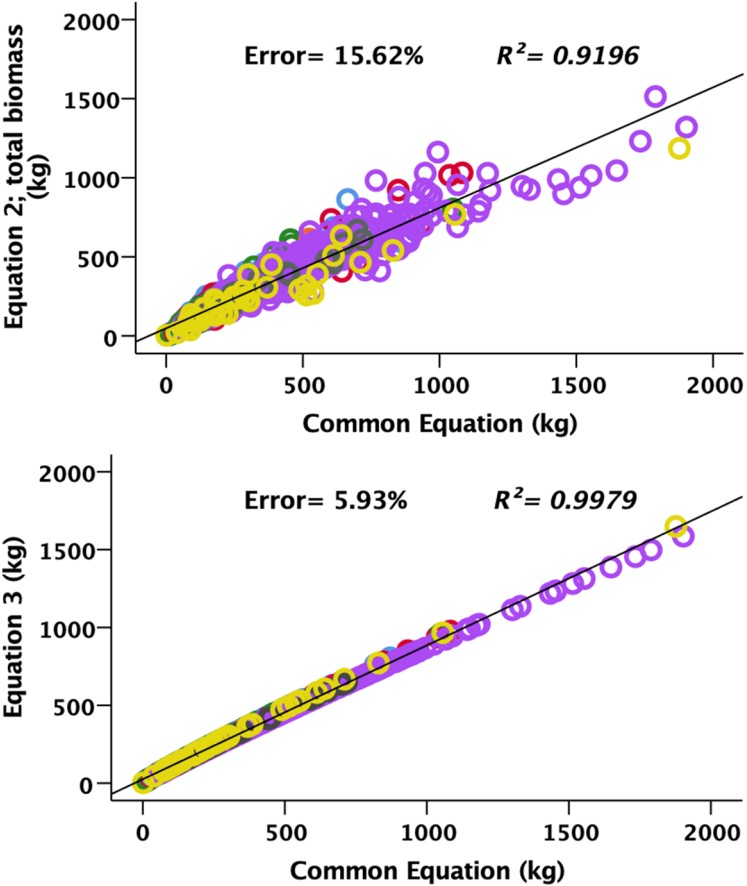

**Figure 4.** Five developed equations of aboveground biomass (kg) estimation of 1000 standing trees regarding species in the 1 ha plot inventory of Sungai Pulai Forest Reserve. Blue: *B. cylindrica*, red: *B. parviflora*; green: *B. sexangula*; orange: *C. tagal*; purple: *R. apiculata*; dark grey: *R. mucronata*; and yellow: *X. granatum*.

The H variable is not a good estimator for biomass estimation, and it produced the highest percentage error (>15%) when implemented in a large data set (Figure 4). Conversely, the combination of D and ρ variables resulted in the largest improvement ($R^2$ values close to one) for aboveground biomass estimation and showed the lowest percentage error (<6%) (Figure 4). These findings align with the earlier observations by Putz and Chan [27] and Ong et al. [26] that D and ρ variables for mangrove species provide a reliable means of estimating aboveground biomass (Equation (3)).

Further regression analysis was performed for the developed equation regarding the species to depict the values of which species were overestimated or underestimated for large-scale data estimation. Figure 5 indicates that the Equation (3) recorded the highest $R^2$ value (0.9943), followed by both equations of the single D variable (Equation (1) $R^2$ = 0.9337; Equation (3) $R^2$ = 0.9355). The inclusion of the H variable for Equation (2) yielded the lowest $R^2$ value (0.8388 and 0.8367, respectively). Equation (2) in Figure 5 showed inconsistent results when the H variable was incorporated. This may be due to the difficulty of stem height measurement in situ, the mangrove soil condition (muddy soil), and the tidal water that always introduce a bias and greater inaccuracy of measurements. Meanwhile, the regression line indicates the equation over-estimated and under-estimated biomass at low observed value and high observed value, respectively (Figure 5).

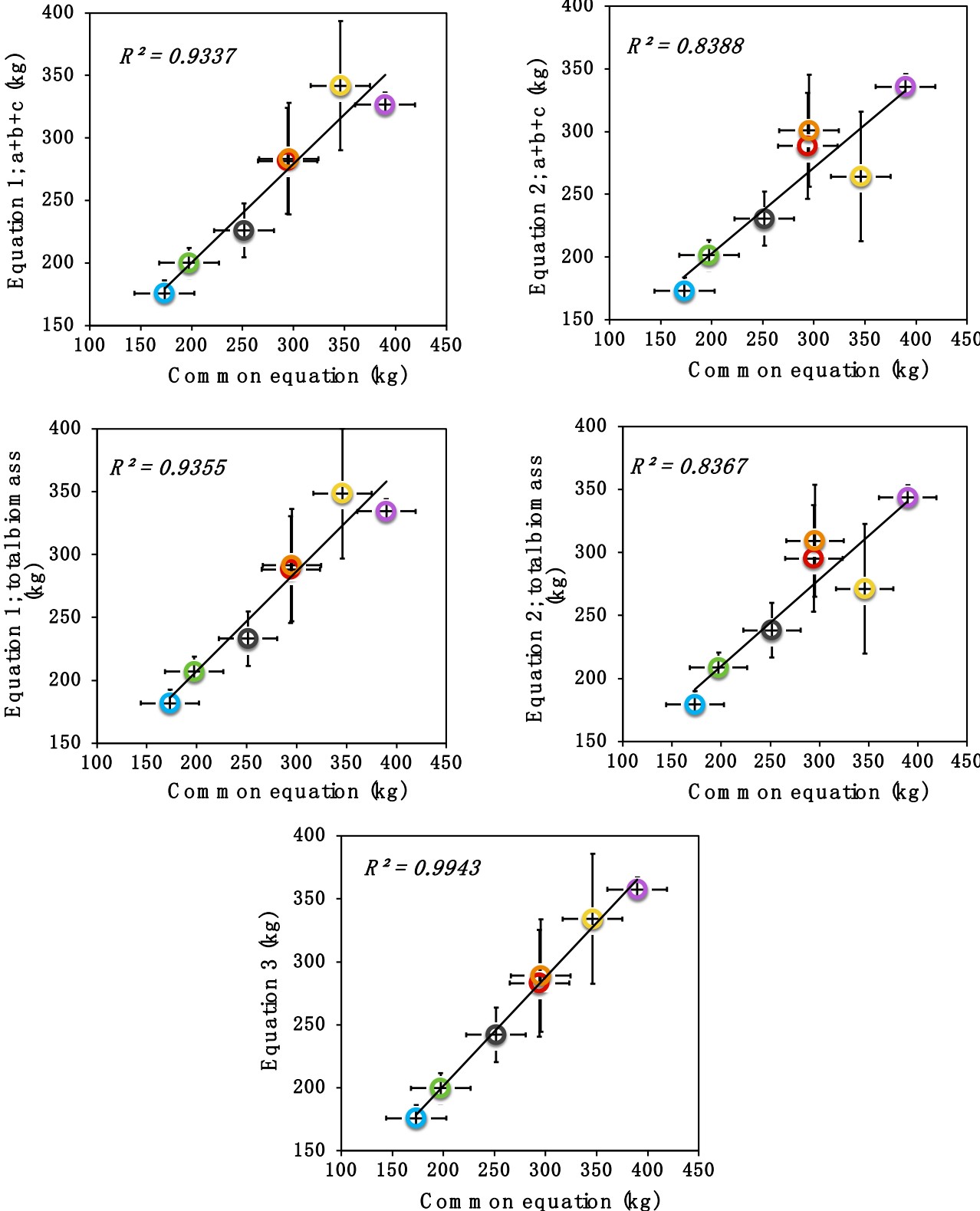

**Figure 5.** Five developed equations of aboveground biomass (kg) estimation of 1000 standing trees regarding species in the 1 ha plot inventory of Sungai Pulai Forest Reserve. Blue: *B. cylindrica*, red: *B. parviflora*; green: *B. sexangula*; orange: *C. tagal*; purple: *R. apiculata*; dark grey: *R. mucronata*; and yellow: *X. granatum*.

The estimation point (species) located under the regression line yielded an under-estimate of the biomass, meanwhile the point (species) located above the line yielded an over-estimate of the biomass. All equations showed both underestimation and over-estimation of the total biomass of the seven species except Equation (3), which yielded well-fitted data for the seven species (Figure 5). Meanwhile, Figure 4 indicates that the biomass estimation was accurately estimated in the large data set regardless of the species by considering the ρ variable. Other equations fitted the data only when the ρ variable was included. In contrast, excluding the ρ variable in the developed equation yields an inaccurate model both for small-scale (15 trees) and large-scale data (1000 trees), as the species sample becomes imbalanced. Thus, the ρ variable is important in reducing the error in the biomass estimation. Therefore, the best-fit equation for all seven species was found for Equation (3). The developed equation of mixed species and uneven age in the study site provides a different successful establishment of the equation (Table 2), as compared to the single species equation [61,62].

## 4. Conclusions

The single variable (D) equation provides an accurate estimation, which is slightly improved when incorporated with the H variable. The exclusion of the variable H might be considered on time consuming grounds and for difficult events, however, both D and H variables show inconsistent results for large-scale data and imbalanced sample species. Meanwhile, the best fit either for small-scale or large scale-data, as well as for imbalanced sample species was achieved following the inclusion of the ρ variable. We suggest that the ρ variable should be considered as an important determinant variable in mixed mangrove species and uneven-aged stand for aboveground biomass estimation. This valuation can both improve and influence decision-making in forest development and conservation.

**Author Contributions:** Conceptualization, H.A.-H., F.-N.M.-I., J.M., Z.S., T.-M.T.-I. and L.-S.M.; methodology, H.A.-H., F.-N.M.-I., R.A., Z.S. and H.-R.N.; data acquisition, H.A.-H., F.-N.M.-I., J.M. and A.-M.J.; formal analysis, H.A.-H. and F.-N.M.-I.; writing-original draft preparation, H.A.-H., F.-N.M.-I. and J.M.; Writing-review and editing, H.A.-H., F.-N.M.-I., J.M., Z.S., R.A., T.-M.T.-I., L.-S.M., A.-M.J. and H.-R.N.; supervision, H.A.-H. All authors have read and agreed to the published version of the manuscript.

**Funding:** This research received no external funding.

**Acknowledgments:** We would like to express our gratitude to the Forestry Department Peninsular Malaysia for the assistance and permission to conduct this study at Sungai Pulai Forest Reserve, Johor, Malaysia.

**Conflicts of Interest:** The authors declare no conflict of interest.

## Appendix A

**Table A1.** List of allometric equations for estimating aboveground biomass (ABG) by using DBH (D), height (H) and wood density (ρ).

| Species Group | Equation | $R^2$ | N | Data Origin | D Max (cm) | Source |
|---|---|---|---|---|---|---|
| General equation | $B = \rho \times \exp[-1.349 + 1.980 \times \ln(D) + 0.207 \times (\ln(D))^2 - 0.0281 \times (\ln(D))^3]$ | unknown | 84 | Americas | 42.0 | Chave et al. [41] |
| General equation | $B = 0.168 \times \rho \times (D)^{2.471}$ | 0.99 | 84 | Americas | 42.0 | Chave et al. [41]; Komiyama et al. [11] |
| General equation | $B = 0.251\,\rho\,(D)^{2.46}$ | 0.98 | 104 | Asia | 49.0 | Komiyama et al. [29] |

**Table A1.** *Cont.*

| Species Group | Equation | $R^2$ | N | Data Origin | D Max (cm) | Source |
|---|---|---|---|---|---|---|
| Specific tree equations—Asia-Pacific region | | | | | | |
| *Rhizophora apiculata* | $B = 0.1709D^{2.516}$ | 0.98 | 20 | Malaysia | 30.0 | Putz & Chan [27] |
| *Rhizophora apiculata* | $B = 0.043D^{2.63}$ | 0.97 | 34 | Indonesia | 40.0 | Amira [63] |
| *Rhizophora apiculata* (wood mass) | $B_{wood} = 0.0695D^{2.644} \times \rho$ | 0.89 | 191 | Micronesia | 60.0 | Modified from Cole et al. [64]; Kauffman & Cole [65] |
| *Xylocapus granatum* | $B = 0.1832D^{2.21}$ | 0.95 | 30 | Indonesia | 41.0 | Tarlan [66] |

**Appendix B**

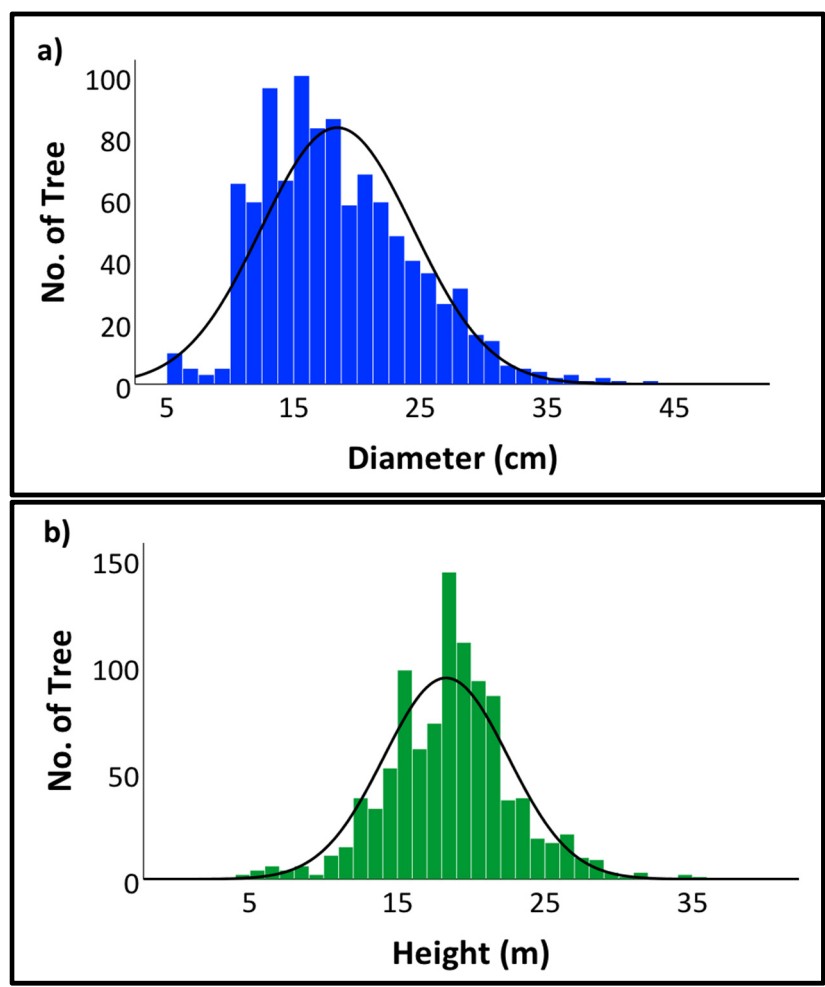

**Figure A1.** Histogram of (**a**) the diameter and (**b**) the height distribution of 1000 standing trees in the 2-ha plot of Sungai Pulai Forest Reserve.

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
