# Peer review of "Allometric Equation for Aboveground Biomass Estimation of Mixed Mature Mangrove Forest"

_forests, doi:10.3390/f13020325_

Round 1
Reviewer 1 Report
Major revision
This manuscript tries to construct robust allometric equations for mixed mature mangrove forests by field surveys, which are important for the accurate estimation of above-ground biomass. The major concern is that the imbalance of different mangrove samples, which may lead to over-optimistic results. For example, R. apiculate species is over half of the total trees with a minor standard deviation. If one species is dominant, it will not be convinced since it cannot be viewed as mixed mangroves. Besides, the result part needs significant improvement in the statement and should be carefully checked. Specifically,
Line 122: As discussed in the manuscript, tree density is an important factor for mangrove biomass estimation. Therefore, the records on tree density from fields surveys should be provided in table 1.
Figure 2: suggest using different colors for different species, which will be more consistent with the topic related to mixed mangroves.
Section 3.2: please carefully check the number for the referenced figure. For example, figure 2 in line 270 should be figure 3.
Line 293-299: cannot find the source of the values of R2, making it hard to follow. Please check thoroughly.
Line 301: without the reference line in figure 3, we cannot conclude that the equation over-estimated biomass at low values and under-estimated at high values.
Author Response
REVIEW REPORT 1
Related to imbalance on different mangrove samples.
Answer/ Justification:
Plot establishments were done near to the river or the estuary. The species distribution dominant may occur for seaward zone and mid-zone of mangrove species, however, gives unbiased to the development of equation due to un-even age of natural regeneration mix species, in which the result depicted that single variable (D) equation gives accurate estimation, and slightly improved when incorporated with H variable. Meanwhile, the inclusion of ρ variable in equation development meanwhile shows the best fit either for small scale or large-scale data, and for imbalance sample regarding mix species.
This is because the ρ was slightly difference in same species, but gives significant difference regarding species and age. Thus ρ inclusion was helps to reduce the error of biomass estimation and efficiently helps in forest management. Furthermore, the number of samples on each species to be harvest is upon the number of trees of each species on primary inventory. Previous studies also showed that R. apiculata was the dominant species in another mangrove forest in Peninsular Malaysia and East Malaysia Thus, ρ variable should be considered as an important determinant variable in mixed mangrove species and uneven-aged stand for aboveground biomass estimation. The inclusion of ρ variable in this equation could also be applied to other mangrove communities with forest characteristics as justify in introduction.
Related to different colours for different species (Figure 2).
Answer/ Justification:
Figure 2 has been corrected to figure 1.
The authors prefer not to differentiate the colour regarding the species.
This is because the analysis in Figure 1 was perfom to study which tree variable/developed equation can fit large-scale data or shows inconsistent result when applied to mix species and regardless species, and the finding depict incorporated of ρ and D in developed equation gives smallest percentage of error when applied to large-scale data (<6%).
Next, the in-depth analysis regarding species was shown in figure 3, in which the developed equation were applied to primary large-scale data in comparison with common equation regarding species, to study which variable/developed equation can fit or shows inconsistency result when perfom to the imbalance large-scale sample regarding the species, and the finding depict incorporated of D and H variable in developed equation gives inconsistent result for imbalance sample regarding species. Meanwhile, the inclusion of ρ variable in equation development meanwhile shows the best fit either for small scale or large-scale data, and for imbalance sample regarding mix species.
Related to reference figure
Answer/ Justification:
The reference figure has been corrected thoroughly
The source of the value R²
Answer/ Justification:
The R² value were not shown in Figure 3. The R² value however were reflected to the regression line of each of the equation, in which Eq. (5) with the highest R² (0.9952) shown the best fit regarding species, meanwhile Eq. (2) adn (4) with the lowest R² (0.8468 and 0.8452, respectively) shown the inconsistent result regarding species.
Over-estimated and under-estimated
Answer/ Justification:
Over-estimated and under-estimated firstly can be show from the R² value. The highest R² value less/none over-estimated and under-estimated regarding species such as in Eq. (5). Another way of to determine the estimation is point (species) that located under-line was under-estimated of the biomass, and point (species) that located upper-line was over-estimated of the biomass in both using common equation and equation developed in this study. Meanwhile, the point (species) located in-line was fit or more accurate for the biomass estimation in both using common equation and equation developed in this study.
Reviewer 2 Report
The manuscript titled "Evaluation Development of Allometric Equation for Aboveground Biomass Estimation of Mixed Mature Mangrove Forest" reported on the development of site-specific allometric equations for estimating above-ground biomass in a mix-mature mangrove forest at Sg. Pulai Forest Reserve, Johor, Malaysia.
The authors measured diameter, and height for 1000 individuals and identified the species for each individual. They further destructively sampled and measures dbh, height, wood density, and biomass by components for 15 individuals belonging to seven species to develop site-specific multi-species allometric equations.
The study is interesting giving its application for accurate estimation of biomass. The study also has implications for REDD+ programmes and any other climate change mitigation or adaptation programme. It further the literature on the estimation of above-ground biomass in a critical ecosystem such as mangroves.
However, there are a number of issues that need to be addressed by the authors before it can be accepted for publications.
- TITLE : The title should be revised to be clearer than the current one. I suggest : "Allometric Equation for Aboveground Biomass Estimation of a Mixed Mature Mangrove Forestin Malaysia"
- GRAMMATICAL ERRORS AND NEED FOR REPHRASING : There are several grammatical errors in the manuscript. There are also many sentences that need to be rephrased. I highlight many of these grammatical errors and sentences to be rephrased in the edited manuscript, but I believe that there remain some in the manuscript. I suggest the manuscript to be revised by a native speaker for consistency in the writing style.
- INTRODUCTION : Although the aim of the study is provided, it would be good to add specific objectives and hypotheses to provide a specific framework for reading the manuscript. The fact that several models were tested provide opportunities for stating some hypotheses.
- MATERIAL AND METHODS
- Line 107 : authors mentionned 1000 trees. How this number was obtained? Did their consider all individuals in each plot or their just select some? If they selected some, then, they should indicate how they did it and why.
- Line 137 : The drying temperature is missing
- Lines 150 – 163 : The presentations of the tested equations is confusing. Better provide explicit form of each equation instead of giving two different denominations to the same equation. Several other comments on this point can be found in the edited manuscript.
- The statistical analyses should also be complemented with the check of the properties of models’ residuals (namely : normality, homoskedascity, independance). The models should also be compared with statistics other than the R square (e.g. AIC, RMSE, etc.)
- Line 180: Authors wrote the following : "… between the observed value and predicted value for an equation using D, D2H and ρ variables". I could not see an equation where D2H was used as explanatory factor. The equation on line 152 does not actually model with D2H. Kindly revise the formula accordingly ang give explicit form for each equation as indicated above. This will avoid confusion. Also see my comment on table 2, Eq. 2 in the edited manuscript.
- RESULTS and DISCUSSION
- In several instances, the authors mention « common equation ». However, the meaning of this expression is not clear. The explanation on line 165 is not clear. Authors are invited to better explain this, since this is central for the second part of the manuscript who focused on the 1000 trees.
- Lines 270-280 : I could not get what has actually been done here. The authors need to explain more what they did to allow the reader to comprehend their manuscript. The text is about 1000 individuals, but the figure 2 it is referring to only show 15 points.
- Lines 293 – 319 : The results presented here is not clear. What analysis has been done here ? How did authors obtain the biomass for the 1000 individuals. I don’t know if I am right, but, I imagine that authors used an existing equation to estimate biomass of the 1000 individuals and then used each of their 5 equations to estimate the biomass for the same 1000 individuals. Using these two values, their made some comparisons. If I am right, then which equation they used ? I could not see that clearly in the manuscript.

Author Response
REVIEW REPORT 2
- Title suggestion to Allometric Equation for Aboveground Biomass Estimation of a Mixed Mature Mangrove Forest in Malaysia
Answer/ Justification:
The title has been changed. The authors however prefer to exclude the country name, as the methods, analysis and the discussions were valid to be refer and apply regardless country region
- Grammatical Error
Answer/ Justification:
The abstract has been improved
The sentence has been rephrase for the whole manuscript
- Introduction
Answer/ Justification:
The introduction has been improved thoroughly regarding the objective and the justification of the study. The term ‘within’ and ‘between’ species also has been changed to ‘regardless’ and ‘regarding’ species to avoid readers confusion
- Material and Methods
Answer/ Justification:
- The purpose in selection of 1000 standing trees from 1386 trees is to make the graph simple, fit and easy for the reader, and 1000 trees is still valid for large-scale data
- The missing temperature has been added
- The authors prefer to fix the equation form as the focus of the study is to explained how much gain in tree variability recorded the biomass accumulation on stem, branch and twigs, and leaves and to avoid reader confusion on various equation form.
- The authors belief the analysis were sufficiently enough:
- 5 equation development
- Predicted value and observed value of 15 destructive sample were applied to the developed equation and common equation
iii. The developed equation were applied to primary large-scale data (1000 standing trees) with percentage of error in comparison with common equation regardless species to study which variable/developed equation can fit with mix species
- The developed equation were applied to primary large-scale data in comparison with common equation regarding species to study which variable/developed equation can fit or shows inconsistency result when perfom to the imbalance large-scale sample regarding the species
- The missing value in Table 2 has been improved. All denomiate of D2H has been change to D²H. Thank you for your concern
- Results and Discussion
Answer/ Justification:
‘common allometric equation’ is a well-known equation that developed by Komiyama et al. (2005) to estimate biomass of mangrove species in South-East Asia. This equation take in considered for comparison purposes, as ‘common allometric equation’ developed from destructive sample:
- Destructively sample in South-East Asia
- The study specific for mangrove forest
iii. The number of destructive sample (104) much higher compared to this study (15), which mean cover broader in tree sizes, thus we need to check either our validity of developed equation since we incorporate with much smaller number of destructive sample.
From the comparison, our finding revealed that small destructive sampling (15 trees) R² (0.9712) with the inclusion of ρ variable provide a reliable and validated equation instead of the common equation (R²=0.9713). Furthermore, depicted that the developed equation in this study may applicable to other mangrove communities with similar geomorphological condition
All the discussion also has been improve in proper data discussion and present.Thank you so much for your concern.
Reviewer 3 Report
Since the structure and functions of mangroves are highly site-specific, the authors' efforts to develop the site-specific allometric equation for the precise estimation of above-ground biomass and carbon stock are highly significant. Certainly, the manuscript deserves publication in "Forest". However, I feel that the following points could be considered for strengthening the information presented in the manuscript.
Major
- Authors compared their allometric equation; in that way, a discussion on “comparison between equation 5 and common equation developed by Komiyama et al (2005)” would be significant; because both equations are from the same region and both use wood density and DBH to estimate the biomass.
- A total of 2ha area has been assessed for this study. So I recommend authors to present estimated biomass and carbon stock of the studied sites (based on equation 5 and common equation of Komiyama et al. 2005)
Minor
Line No: 137 temperatures is missing after “15 days at”
Line No. 251-254 change font of species names in italics
Author Response
REVIEW REPORT 3
The whole manuscript
Answer/ Justification:
- The title has been changed. The authors however prefer to exclude the country name, as the methods, analysis and the discussions were valid to be refer and apply regardless country region
- The abstract has been improved. The sentence has been rephrase for the whole manuscript
iii. The term ‘within’ and ‘between’ species has been change to ‘regardless’ and ‘regarding’ species to avoid readers confusion
- The introduction has been improved regarding the objective and the justification of the study
- The missing temperature has been added
- The name of tree species has been change in Italics
vii. The authors belief the analysis were sufficiently enough:
- 5 equation development
- Predicted value and observed value of 15 destructive sample were applied to the developed equation and common equation
- The developed equation were applied to primary large-scale data (1000 standing trees) with percentage of error in comparison with common equation regardless species to study which variable/developed equation can fit with mix species
- The developed equation were applied to primary large-scale data in comparison with common equation regarding species to study which variable/developed equation can fit or shows inconsistency result when perfom to the imbalance large-scale sample regarding the species
However, the suggestion was nice and gives authors idea in drafting other manuscript in future by considering primary data inventory from other mangrove forest. Thank you in advance.
Reviewer 4 Report
In this paper, the field work is sufficient, and the woody density is an important factor for estimating mangrove biomass, and the model for estimating mangrove biomass is given. But there are still some problems with this paper, as shown below. Based on these opinions, I think this paper needs a major revision.
- There is no clear research objective and content in the introduction
- Since there are models for estimating the biomass of single species, why do we need models for estimating the biomass of mixed species? In the introduction, I didn't read the narrative. Moreover, in the subsequent application, remote sensing could not obtain the wood density information of ground mangrove mixed tree species, and the model could not be applied to remote sensing. In the study of UAV extraction of mangrove biomass, tree species and stand information can be directly distinguished, and the biomass estimation model of single tree species is obviously more reliable. So, what is the practical significance of the model established in this study?
- The name of FIG. 4 is not illustrative, please reconsider. Units and legends are missing, please add them.
- There may be differences in wood density, height and DBH among different species, and some species occupy a small proportion in this plot and contribute little to model fitting. Then, is the general model applicable to this plot applicable to other mangrove communities applicable? I noticed in Figure 4 that the species that were poorly adapted to the model were the ones that were less abundant in the plot.
- Please pay attention to the logic and coherence of the description in the conclusion and discussion section. In addition, there are many case, superscript, and formatting errors in the full text, please modify. For example, the square symbol in lines 230,236,242.
Author Response
REVIEW REPORT 4
Allometric relationships between aboveground biomass and DBH parameter have been reported for specific mangrove species such as Rhizophora apiculata and Bruguiera parviflora. In this study, the priority focused is in determining biomass accumulation on natural mature mangrove forest that occupied on mixed species to help understand more on forest conditions that located facing other land areas (Singapore) or mangrove islands where it possibly receives seed sources from the nearest forest, and not facing directly to the vast ocean that facing rough tides, huge waves, strong winds, and tropical storms such as typhoons and hurricanes. By using primary data, the verified single develops equation (with several relationship) that can fits the available mangrove species could help in managing forest efficiently and generally providing the relevant figures of forest canopy layer as well as forest profile regarding biomass accumulation and carbon stock at this particular forest area, instead of applying single species models that may need higher cost and time consuming in data inventory, gathering, and presenting.
In comparison with ‘common equation’ that conducted on the other mangrove in South-East Asia with much higher in destructive sample (104 trees), revealed that small destructive sampling (15 trees) R² (0.9712) in this study with the inclusion of ρ variable provide a reliable and validated equation instead of the common equation (R²=0.9713), thus depicted that the developed equation in this study may applicable to other mangrove communities with similar geomorphological condition. Furthermore, the inclusion of ρ variable in equation development shows the best fit either for small scale or large-scale data, and for imbalance sample regarding species. We suggest that ρ variable should be considered as an important determinant variable in mixed mangrove species and uneven-aged stand for aboveground biomass estimation. The developed equation of mixed species and uneven age in study site provide the different successful establishment of the equation, as compared to single species equation. This valuation can improve and influence decision-making in forest development and conservation.
Unmanned Aerial Vehicle (UAV) such as drone and remote sensing data in capturing forest imagery based on forest canopy that further described based on the colour features gives advantages, particularly the forest areas that are hard to access in managing and monitoring forest from undesirable things such as illegal logging, however unreliable to estimate wood density and biomass by tree component (trunk, branch, leaves, flowers, and fruits) on the ground that support the aboveground biomass accumulation, in which these data were then are crucial in determining forest carbon stock. Different definition of forest canopy height (such as the mean height of all trees, basal-area weighted height or height of the tallest tree within a certain area) that generated from airborne laser/ Lidar (ALS) may lead to differing results from the same datasets, [27–29]. Furthermore, mangroves unique structure of aboveground root systems and as regular inundation might result in the error in the height retrievals In addition, the developed equation may explained how much gain in tree variability recorded the biomass accumulation on stem, branch and twigs, and leaves thus gives an information Thus, the proper and accurate estimation of forest biomass is one of the 22 elements to determine Total Economic Valuation (TEV) for forest ecosystem and services based on the framework by The Economics of Ecosystem and Biodiversity (TEEB) 2008 and De Groot et al (2012).
Round 2
Reviewer 1 Report
Major revision
This manuscript tries to construct robust allometric equations for mixed mature mangrove forests by field surveys, which are important for the accurate estimation of above-ground biomass. The major concern is the imbalance of different mangrove samples, which may lead to over-optimistic results. For example, R. apiculate species is over half of the total trees with a minor standard deviation. If one species is dominant, it will not be convinced since it cannot be viewed as mixed mangroves. Besides, the result part needs significant improvement in the statement and should be carefully checked. Specifically,
Line 122: As discussed in the manuscript, tree density is an important factor for mangrove biomass estimation. Therefore, the records on tree density from fields surveys should be provided in table 1.
Figure 2: suggest using different colors for different species, which will be more consistent with the topic related to mixed mangroves.
Section 3.2: please carefully check the number for the referenced figure. For example, figure 2 in line 270 should be figure 3.
Line 293-299: cannot find the source of the values of R2, making it hard to follow. Please check thoroughly.
Line 301: without the reference line in figure 3, we cannot conclude that the equation over-estimated biomass at low values and under-estimated at high values.
Author Response
REVIEWER 1
This manuscript tries to construct robust allometric equations for mixed mature mangrove forests by field surveys, which are important for the accurate estimation of above-ground biomass. The major concern is the imbalance of different mangrove samples, which may lead to over-optimistic results. For example, R. apiculate species is over half of the total trees with a minor standard deviation. If one species is dominant, it will not be convinced since it cannot be viewed as mixed mangroves. Besides, the result part needs significant improvement in the statement and should be carefully checked. Specifically,
Comment: Line 122: As discussed in the manuscript, tree density is an important factor for mangrove biomass estimation. Therefore, the records on tree density from fields surveys should be provided in table 1.
Comments: Figure 2: suggest using different colors for different species, which will be more consistent with the topic related to mixed mangroves.
Comments: Section 3.2: please carefully check the number for the referenced figure. For example, figure 2 in line 270 should be figure 3.
Answer: The reference figure has been corrected thoroughly
Comment: Line 293-299: cannot find the source of the values of R2, making it hard to follow. Please check thoroughly.
Comment: Line 301: without the reference line in figure 3, we cannot conclude that the equation over-estimated biomass at low values and under-estimated at high values.
Reviewer 2 Report
The authors satisfactorily addressed most of my comments. But I am not satisfied with the response to the comments on statistical analyses.
I am afraid, I do not share the view that “The analyses were sufficiently enough”. My comments on statistical analyses were not really to do additional analyses, but rather to check properties of the residuals of the models established. Simply estimating equations (i.e., estimating coefficients) is not enough. The properties of the residuals of the equations need to be checked. This is classic when it comes to linear models, especially considering the fact that only 15 individuals (small sample) were considered for establishing the equations. Checking residuals normality, homoskedasticity, and independence is a requirement for validating the models.
There are still some grammatical errors which authors need to carefully check. I am not sure if the manuscript has been proof-read by a native English speaker.
The so-called ‘common allometric equation’ has been better clarified. Authors should bear in mind that there are several other equations that can also be considered as “common allometric equation’. So, it is crucial to give details as much as possible for the reader. I am OK with the current version.
Author Response
REVIEWER 2
The authors satisfactorily addressed most of my comments. But I am not satisfied with the response to the comments on statistical analyses.
I am afraid, I do not share the view that “The analyses were sufficiently enough”. My comments on statistical analyses were not really to do additional analyses, but rather to check properties of the residuals of the models established. Simply estimating equations (i.e., estimating coefficients) is not enough. The properties of the residuals of the equations need to be checked. This is classic when it comes to linear models, especially considering the fact that only 15 individuals (small sample) were considered for establishing the equations. Checking residuals normality, homoskedasticity, and independence is a requirement for validating the models.
There are still some grammatical errors which authors need to carefully check. I am not sure if the manuscript has been proof-read by a native English speaker.
The so-called ‘common allometric equation’ has been better clarified. Authors should bear in mind that there are several other equations that can also be considered as “common allometric equation’. So, it is crucial to give details as much as possible for the reader. I am OK with the current version.
Answer:
We have improved the whole manuscript. The abstract has been changed